# Relevant sparse codes with variational information bottleneck

**Matthew Chalk**
IST Austria
Am Campus 1
A - 3400 Klosterneuburg, Austria

**Olivier Marre**
Institut de la Vision
17, Rue Moreau
75012, Paris, France

**Gasper Tkacik**
IST Austria
Am Campus 1
A - 3400 Klosterneuburg, Austria

## Abstract

In many applications, it is desirable to extract only the relevant aspects of data. A principled way to do this is the information bottleneck (IB) method, where one seeks a code that maximizes information about a 'relevance' variable, $Y$, while constraining the information encoded about the original data, $X$. Unfortunately however, the IB method is computationally demanding when data are high-dimensional and/or non-gaussian. Here we propose an approximate variational scheme for maximizing a lower bound on the IB objective, analogous to variational EM. Using this method, we derive an IB algorithm to recover features that are both relevant and sparse. Finally, we demonstrate how kernelized versions of the algorithm can be used to address a broad range of problems with non-linear relation between $X$ and $Y$.

## 1 Introduction

An important problem, for both humans and machines, is to extract relevant information from complex data. To do so, one must be able to define which aspects of data are relevant and which should be discarded. The 'information bottleneck' (IB) approach, developed by Tishby and colleagues [1], provides a principled way to approach this problem. The idea behind the IB approach is to use additional 'variables of interest' to determine which aspects of a signal are relevant. For example, for speech signals, variables of interest could be the words being pronounced, or alternatively, the speaker identity. One then seeks a coding scheme that retains maximal information about these variables of interest, constrained on the information encoded about the input.

The IB approach has been used to tackle a wide variety of problems, including filtering, prediction and learning [2-5]. However, it quickly becomes intractable with high-dimensional and/or non-gaussian data. Consequently, previous research has primarily focussed on tractable cases, where the data comprises a countably small number of discrete states [1-5], or is gaussian [6].

Here, we extend the IB algorithm of Tishby et al. [1] using a variational approximation. The algorithm maximizes a lower bound on the IB objective function, and is closely related to variational EM. Using this approach, we derive an IB algorithm that can be effectively applied to 'sparse' data in which input and relevance variables are generated by sparsely occurring latent features. The resulting solutions share many properties with previous sparse coding models, used to model early sensory processing [7]. However, unlike these sparse coding models, the learned representation depends on: (i) the relation between the input and variable of interest; (ii) the trade-off between encoding quality and compression. Finally, we present a kernelized version of the algorithm, that can be applied to a large range of problems with non-linear relation between the input data and variables of interest.

## 2 Variational IB

Let us define an input variable $X$, as well as a 'relevance variable', $Y$, with joint distribution $p(y, x)$. The goal of the IB approach is to compress the variable $X$ through another variable $R$, while conserving information about $Y$. Mathematically, we seek an encoding model, $p(r|x)$, that maximizes:

$$
\begin{aligned}
L_{p(r|x)} &= I(R;Y) - \gamma I(R;X) \\
&\equiv \langle \log p(y|r) - \log p(y) + \gamma \log p(r) - \gamma \log p(r|x) \rangle_{p(r,x,y)},
\end{aligned}
\tag{1}
$$

where $0 < \gamma < 1$ is a Lagrange multiplier that determines the strength of the bottleneck.

Tishby and colleagues showed that the IB loss function can be optimized by applying iterative updates: $p_{t+1}(r|x) \propto p_t(r) \exp\left[-\frac{1}{\gamma} \int_y p(y|x) \log \frac{p(y|x)}{p_t(y|r)}\right]$, $p_{t+1}(r) = \int_x p(x) p_{t+1}(r|x)$ and $p_{t+1}(y|r) = \int_x p(y|x) p_{t+1}(x|r)$ [1]. Unfortunately however, when $p(x, y)$ is high-dimensional and/or non-gaussian these updates become intractable, and approximations are required.

Due to the positivity of the KL divergence, we can write, $\langle \log q(\cdot) \rangle_{p(\cdot)} \leq \langle \log p(\cdot) \rangle_{p(\cdot)}$ for any approximative distribution $q(\cdot)$. This allows us to formulate a variational lower bound for the IB objective function:

$$
\begin{aligned}
\tilde{L}_{p(r|x),q(y|r),q(r)} &= \frac{1}{N} \sum_{n=1}^{N} \langle \log q(y_n|r) + \gamma \log q(r) - \gamma \log p(r|x_n) \rangle_{p(r|x_n)} \\
&\leq L_{p(r|x)},
\end{aligned}
\tag{2}
$$

where $q(y_n|r)$ and $q(r)$ are variational distributions, and we have replaced the expectation over $p(x, y)$ with the empirical expectation over training data. (Note that, for notational simplicity we have also omitted the constant term, $H_Y = -\langle \log p(y) \rangle_{p(y)}$.)

Setting $q(y_n|r) \leftarrow p(y_n|r)$ and $q(r) \leftarrow p(r)$ fully tightens the bound (so that $\tilde{L} = L$), and leads to the iterative algorithm of Tishby et al. However, when these exact updates are not possible, one can instead choose a restricted class of distributions $q(y|r) \in Q_{y|r}$ and $q(r) \in Q_r$ for which inference is tractable. Thus, to maximize $\tilde{L}$ with respect to parameters $\Theta$ of the encoding distribution $p(r|x, \Theta)$, we repeat the following steps until convergence:

- For fixed $\Theta$, find $\{q^{new}(y|r), q^{new}(r)\} = \arg\max_{\{q(y|r),q(r)\} \in \{Q_{y|r}, Q_r\}} \tilde{L}$

- For fixed $q(y|r)$ and $q(r)$, find $\Theta = \arg\max_{\Theta} \tilde{L}$.

We note that using a simple approximation for the decoding distribution, $q(y|r)$, can carry additional benefits, besides rendering the IB algorithm tractable. Specifically, while an advantage of mutual information is its generality, in certain cases this can also be a drawback. That is, because Shannon information does not make any assumptions about the code, it is not always apparent how information should be best extracted from the responses: just because information is 'there' does not mean we know how to get at it.

In contrast, using a simple approximation for the decoding distribution, $q(y|r)$ (e.g. linear gaussian), constrains the IB algorithm to find solutions where information about $Y$ can be easily extracted from the responses (e.g. via linear regression).

## 3 Sparse IB

In previous work on gaussian IB [6], responses were equal to a linear projection of the input, plus noise: $r = Wx + \eta$, where $W$ is an $N_r \times N_x$ matrix of encoding weights, and $\eta \sim \mathcal{N}(\eta|0, \Sigma)$, where $\Sigma$ is an $N_r \times N_r$ covariance matrix. When the joint distribution, $p(x, y)$, is gaussian, it follows that the marginal and decoding distributions, $p(r)$ and $p(y|r)$, are also gaussian, and the parameters of the encoding distribution, $W$ and $\Sigma$, can be found analytically.

To illustrate the capabilities of the variational algorithm, while permitting comparison to gaussian IB, we begin by adding a single degree of complexity. In common with gaussian IB, we consider

a linear gaussian encoder, $p(r|x) = \mathcal{N}(r|Wx, \Sigma)$, and decoder, $q(y|r) = \mathcal{N}(y|Ur, \Lambda)$. However, unlike gaussian IB, we use a student-t distribution to approximate the response marginal: $q(r) = \prod_i \text{Student}\left(r_i|0, \omega_i^2, \nu_i\right)$, with scale and shape parameters, $\omega_i^2$ and $\nu_i$, respectively. When the shape parameter, $\nu_i$, is small then the student-t distribution is heavy-tailed, or 'sparse', compared to a gaussian distribution. Thus, we call the resulting algorithm 'sparse IB'. Unlike gaussian IB, the introduction of a student-t marginal means the IB algorithm cannot be solved analytically, and one requires approximations.

## 3.1 Iterative algorithm

Recall that the IB objective function consists of two terms: $I(R; Y)$, and $I(R; X)$. We begin by describing how to optimize the lower and upper bound of each of these two terms with respect to the variational distributions $q(y|r)$ and $q(r)$, respectively.

The first term of the IB objective function is bounded from below by:

$$I(R; Y) \geq -\frac{1}{2}\log|\Lambda| - \frac{1}{2N}\sum_n \left\langle (y_n - Ur)^T \Lambda^{-1}(y_n - Ur)\right\rangle_{p(r|x_n)} + \text{const.} \quad (3)$$

Maximizing the lower bound on $I(R; Y)$ with respect to the decoding parameters, $U$ and $\Lambda$, gives:

$$\Lambda = C_{yy} - UWC_{xy}, \quad U = C_{xy}^T W^T \left(WC_{xx}W^T + \Sigma\right)^{-1} \quad (4)$$

where $C_{yy} = \frac{1}{N}\sum_n y_n y_n^T$, $C_{xy} = \frac{1}{N}\sum_n x_n y_n^T$, and $C_{xx} = \frac{1}{N}\sum_n x_n x_n^T$.

Unfortunately, it is not straightforward to express the bound on $I(R; X)$ in closed form. Instead, we use an additional variational approximation, utilising the fact that the student-t distribution can be expressed as an infinite mixture of gaussians: $\text{Student}\left(r|0, \omega^2, \nu\right) = \int_\eta \mathcal{N}\left(r|0, \omega^2\right) \text{Gamma}\left(\eta|\frac{\nu}{2}, \frac{\nu}{2}\right)$ [8]. Following a standard EM procedure [9], one can thus write a tractable lower bound on the log-likelihood, $l \equiv \log\left[\text{Student}\left(r|0, \omega^2, \nu\right)\right]$, which corresponds to an upper-bound on the bottleneck term:

$$I(R; X) \leq \sum_{i,n} \left\langle -\log q(r_i) + \log p(r_i|x_n)\right\rangle_{p(r_i|x_n)} \quad (5)$$

$$\leq \sum_i \left[\frac{1}{2}\log\omega_i^2 + \frac{1}{2N\omega_i^2}\sum_{n=1}^N \xi_{ni}\left\langle r_{ni}^2\right\rangle + f(\nu_i, \xi_i, a_i)\right] - \frac{1}{2}\log|\Sigma| + \text{const.}$$

where $\xi_{ni}$, and $a_i$ denote variational parameters for the $i^{th}$ unit and $n^{th}$ data instance. We used the shorthand notation, $\left\langle r_{ni}^2\right\rangle = w_i x_n x_n^T w_i^T + \sigma_i^2$, where $\sigma_i^2$ is the $i^{th}$ diagonal element of $\Sigma$ and $w_i$ is the $i^{th}$ row of $W$. For notational simplicity, terms that do not depend on the encoding parameters were pushed into the function, $f(\nu_i, \xi_i, a_i)^1$.

Minimizing the upper bound on $I(R; X)$ with respect to $\omega_i^2$, $\xi_{ni}$ and $a_i$ (for fixed $\nu_i$) gives:

$$\omega_i^2 = \frac{1}{N}\sum_{n=1}^N \xi_{ni}\left\langle r_{ni}^2\right\rangle, \quad \xi_{ni} = \frac{\nu_i + 1}{\nu_i + \left\langle r_{ni}^2\right\rangle/\omega_i^2}, \quad a_i = \frac{1}{2}(\nu_i + 1), \quad (6)$$

The shape parameter, $\nu_i$, is then found numerically on each iteration (for fixed $\xi_{ni}$ and $a_i$), by solving:

$$\psi\left(\frac{\nu_i}{2}\right) - \log\left(\frac{\nu_i}{2}\right) = 1 + \frac{1}{N}\sum_{n=1}^N \left[\psi(a_i) - \log\frac{a_i}{\xi_{ni}} - \xi_{ni}\right], \quad (7)$$

where $\psi(\cdot)$ is the digamma function [9].

Next we maximize the full variational objective function $\tilde{L}$ with respect to the encoding distribution, $p(r|x)$ (for fixed $q(y|r)$ and $q(r)$). Maximizing $\tilde{L}$ with respect to the encoding noise covariance, $\Sigma$, gives:

$$\Sigma^{-1} = \frac{1}{\gamma}U^T\Lambda^{-1}U + \frac{1}{N}\Omega^{-1}\sum_{n=1}^N \Xi_n, \quad (8)$$

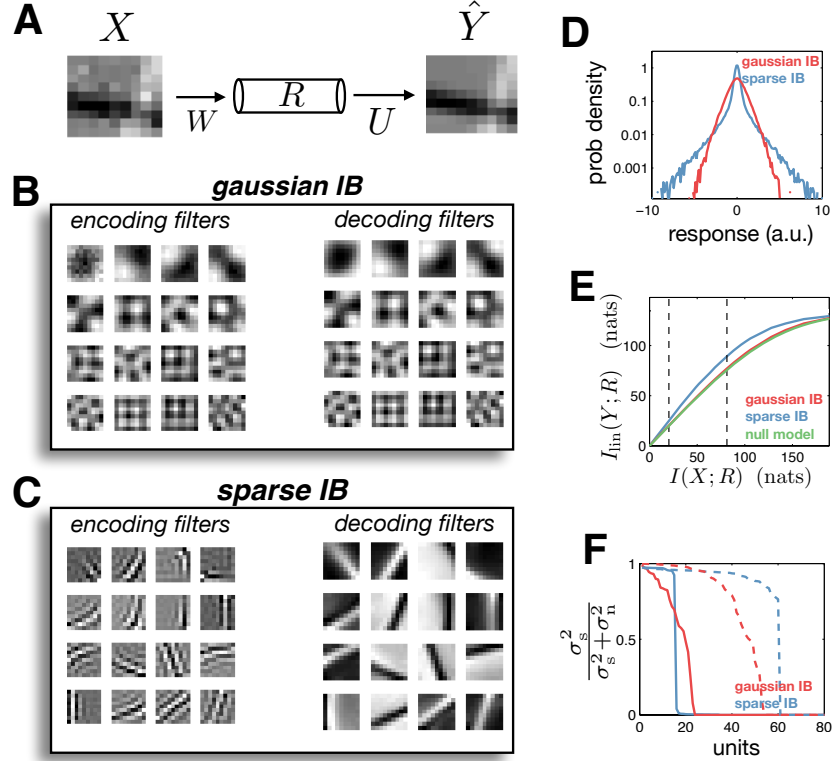

Figure 1: Behaviour of sparse IB and gaussian IB algorithms, on denoising task. (A) Artificial image patches were constructed from combinations of orientated edge-like features. Patches were corrupted with white noise to generate the input, $X$. The goal of the IB algorithm is to learn a linear code that maximized information about the original patches, $Y$, constrained on information encoded about the input, $X$. (B) A selection of linear encoding (left), and decoding (right) filters obtained with the gaussian IB algorithm. (C) Same as B, but for the sparse IB algorithm. (D) Response histograms for the 10 units with highest variance, for the gaussian (red) and sparse (blue) IB algorithms. (E) Information curves for the gaussian (red) and sparse (blue) algorithms, alongside a 'null' model, where responses were equal to the original input, plus white noise. (F) Fraction of response variance attributed to signal fluctuations, for each unit. Solid and dashed curves correspond to strong and weak bottlenecks, respectively (corresponding to the vertical dashed lines in panel E).

where $\Omega$ and $\Xi_n$ are $N_r \times N_r$ diagonal covariance matrices with diagonal elements $\Omega_{ii} = \omega_i^2$, and $(\Xi_n)_{ii} = \xi_{ni}$, respectively.

Finally, taking the derivative of $\tilde{L}$ with respect to the encoding weights, $W$, gives:

$$\frac{\partial \tilde{L}}{\partial W} = U^T \Lambda^{-1} C_{xy}^T - U^T \Lambda^{-1} U W C_{xx} - \gamma \Omega^{-1} \frac{1}{N} \sum_n \Xi_n W x_n x_n^T, \qquad (9)$$

Setting the derivative to zero, we can solve for $W$ directly. One may verify that, when variational parameters, $\xi_{ni}$, are unity, the above iterative updates are identical to the iterative gaussian IB algorithm described in [6].

## 3.2 Simulations

In our framework, the approximation of the response marginal, $q(r)$, plays an analogous role to the prior distribution in a probabilistic generative model. Thus, we hypothesized that a sparse approximation for the response marginal, $q(r)$, would permit the IB algorithm to recover sparsely occurring input features, analogous to the effect of using a sparse prior.

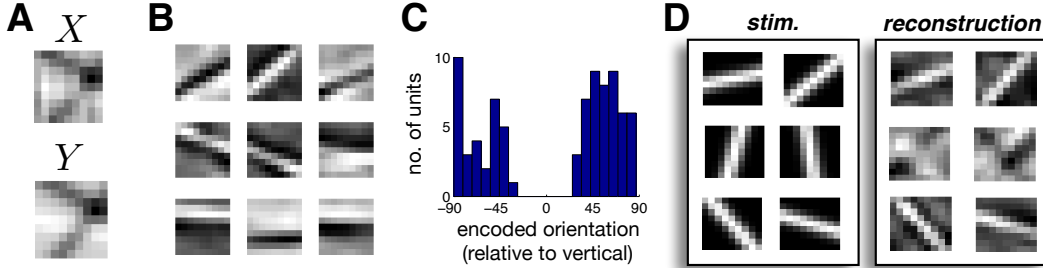

Figure 2: Variant of the task in figure 1, in which the input noise is spatially correlated. (A) Example input $X$ and patch, $Y$. Spatial noise correlations were aligned along the vertical direction. (B) Subset of decoding filters obtained with the sparse IB algorithm. (C) Distribution of encoded orientations. (D) Example stimulus (left) and reconstruction (right) of bars presented at variable orientations (presented with zero input noise, so that $X \equiv Y$ for this example).

To show this, we constructed artificial $9 \times 9$ image patches from combinations of orientated bar features. Each bar had a gaussian cross-section, with maximum amplitude drawn from a standard normal distribution of width 1.2 pixels. Patches were constructed by linearly combining 3 bars, with uniformly random orientation and position.

Initially, we considered a simple de-noising task, where the input, $X$, was a noisy version of the original image patches (gaussian noise, with variance $\sigma^2 = 0.005$; figure 1A). Training data consisted of 10,000 patches. Figure 1B and 1C show a selection of encoding ($W$) and decoding ($U$) filters obtained with the gaussian and sparse IB models, respectively. As predicted, only the sparse IB model was able to recover the original bar features. In addition, response histograms were considerably more heavy-tailed for the sparse IB model (fig. 1D).

The relevant information, $I(R; Y)$, encoded by the sparse model was greater than for the gaussian model, over a range of bottleneck strengths (fig. 1E). While the difference may appear small, it is consistent with work showing that sparse coding models achieve only a small improvement in log-likelihood for natural image patches [10]. We also plotted the information curve for a 'null model', with responses sampled from $p(r|x) = \mathcal{N}(r|x, \sigma^2 I)$. Interestingly, the performance of this null model was almost identical to the gaussian IB model.

Figure 1F plots the fraction of response variance due to the signal, for each unit ($\frac{w_i C_{xx} w_i^T}{w_i C_{xx} w_i^T + \sigma_i^2}$). Solid and dashed curves denote strong and weak bottlenecks, respectively. In both cases, the gaussian model gave a smooth spectrum of response magnitudes, while the sparse model was more 'all-or-nothing'.

One way the sparse IB algorithm differs qualitatively from traditional sparse coding algorithms, is that the learned representation depends on the relation between $X$ and $Y$, rather than just the input statistics. To illustrate this, we conducted simulations with patches corrupted by spatially correlated noise, aligned along the vertical direction (fig. 2A). The spatial covariance of the noise was described by a gaussian envelope, with standard deviation 3 pixels in the vertical direction and 1 pixel in horizontal direction.

Figure 2B shows a selection of decoding filters obtained from the sparse IB model, with correlated input noise. The shape of individual filters was qualitatively similar to those obtained with uncorrelated noise (fig. 1C). However, with this stimulus, the IB model avoided 'wasting' bits by representing features co-orientated with the noise (fig. 2C). Consequently, it was not possible to reconstruct vertical bars from the responses, when bars were presented alone, even with zero noise (fig. 2D).

## 4   Kernel IB

One way to improve the IB algorithm is to consider non-linear encoders. A general choice is: $p(r|x) = \mathcal{N}(r|W\phi(x), \Sigma)$, where $\phi(x)$ is an embedding to a high-dimensional non-linear feature space.

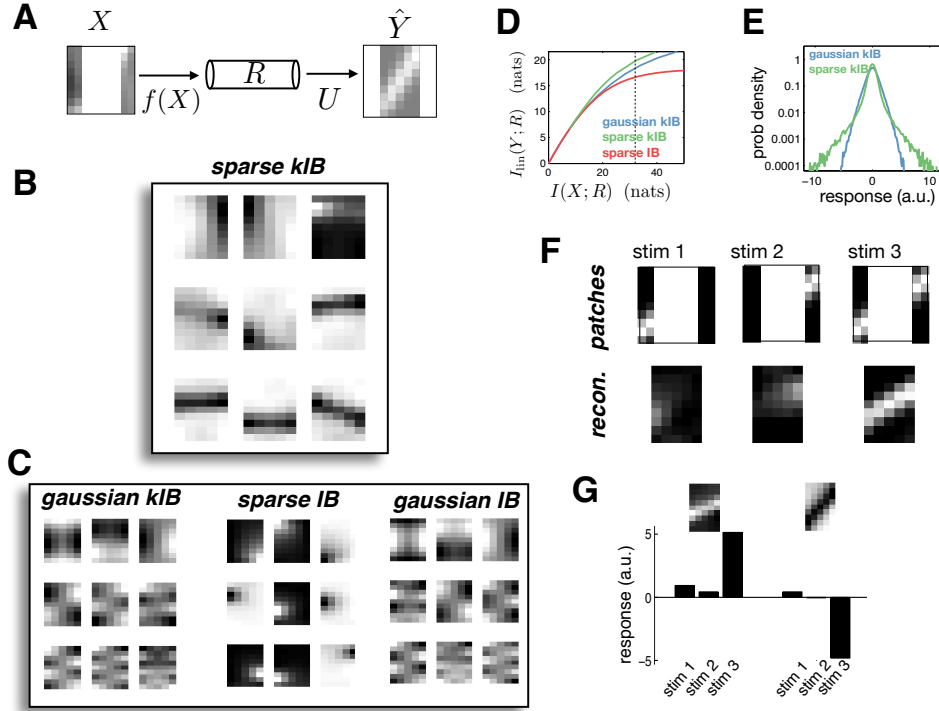

Figure 3: Behaviour of kernel IB algorithm on occlusion task. (A) Image patches were the same as for figure 1. However, the input, $X$, was restricted to 2 columns to either side of the patch. The target variable, $Y$, was the central region. (B) Subset of decoding filters, $U$, for the sparse kernel IB ('sparse kIB') algorithm. (C) As for B, for other versions of the IB algorithm. (D) Information curves for the gaussian kIB (blue) sparse kIB (green) and sparse IB algorithms (red). The bottleneck strength for the other panels in this figure is indicated by a vertical dashed line. (E) Response histogram for the 10 units with highest variance, for the gaussian and sparse kIB models. (F) (above) Three test stimuli, used to demonstrate the non-linear properties of the sparse KIB code. (below) Reconstruction obtained from responses to test stimulus. (G) Responses of two units which showed strong responses to stimulus 3. The decoding filters for these units are shown above the bar plots.

The variational objective functions for both gaussian and sparse IB algorithms are quadratic in the responses, and thus can be expressed in terms of dot products of the row vector, $\phi(x)$. Consequently, every solution for $w_i$ can be expressed as an expansion of mapped training data, $w_i = \sum_{n=1}^{N} a_{in}\phi(x_n)$ [11]. It follows that the variational IB algorithm can be expressed in 'dual space', with responses to the $n^{th}$ input drawn from $r \sim \mathcal{N}(r|Ak_n, \Sigma)$, where $A$ is an $N_r \times N$ matrix of expansion coefficients, and $k_n$ is the $n^{th}$ column of the $N \times N$ kernel-gram matrix, $K$, with elements $K_{nm} = \phi(x_n)\phi(x_m)^T$. In this formulation, the problem of finding the linear encoding weights, $W$, is replaced by finding the expansion coefficients, $A$.

The advantage of expressing the algorithm in the dual space is that we never have to deal with $\phi(x)$ directly, so are free to consider high- (or even infinite) dimensional feature spaces. However, without additional constraints on the expansion coefficients, $A$, the IB algorithm becomes degenerate (i.e. the solutions are independent of the input, $X$). A standard way to deal with this is to add an L2 regularization term that favours solutions with small expansion coefficients. Here, this is achieved here by replacing $\phi_n^T\phi_n$ with $\phi_n^T\phi_n + \lambda I$, where $\lambda$ is a fixed regularization parameter. Doing so, the derivative of $\tilde{L}$ with respect to $A$ becomes:

$$\frac{\partial \tilde{L}}{\partial A} = U^T\Lambda^{-1}YK - \sum_n \left(U^T\Lambda^{-1}U + \gamma\Omega^{-1}\Xi_n\right)A\left(k_nk_n^T + \lambda K\right) \tag{10}$$

Setting the derivative to zero and solving for $A$ directly requires inverting an $NN_r \times NN_r$ matrix, which is expensive. Instead, one can use an iterative solver (we used the conjugate gradients squared

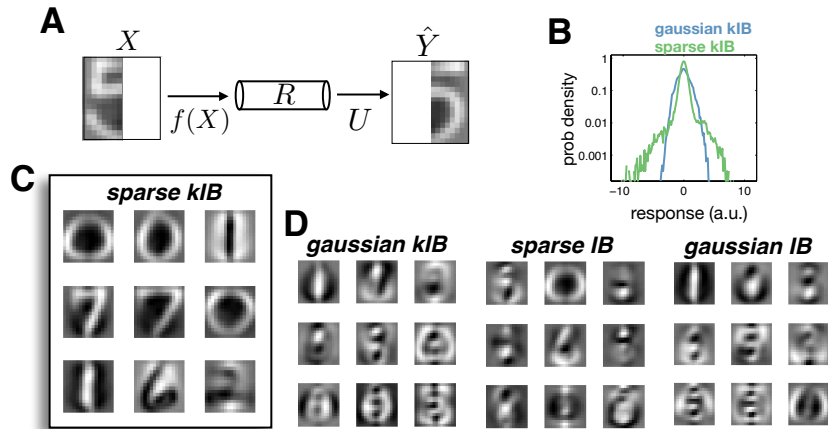

Figure 4: Behaviour of kernel IB algorithm on handwritten digit data. (A) As with figure 4, we considered an occlusion task. This time, units were provided with the left hand side of the image patch, and had to reconstruct the right hand side. (B) Response distribution for 10 neurons with highest variance, for the gaussian (blue) and sparse (green) kIB algorithms. (C) Decoding filters for a subset of units, obtained with the sparse kIB algorithm. Note that, for clearer visualization, we show here the decoding filter for the entire image patch, not just the occluded region. (D) A selection of decoding filters obtained with the alternative IB algorithms.

method). In addition, the computational complexity can be reduced by restricting the solution to lie on a subspace of training instances, such that, $w_i = \sum_{n=1}^{M} a_{in} \phi(x_n)$, where $M < N$. The derivation does not change, only now $K$ has dimensions $M \times N$ [11].

When $q(r)$ is gaussian (equivalent to setting $\Xi_n = I$), solving for $A$ gives:

$$A = \left(U^T \Lambda^{-1} U + \gamma \Omega^{-1}\right)^{-1} U^T \Lambda^{-1} A_{KRR} \qquad (11)$$

where $A_{KRR} = Y(K + \lambda I)^{-1}$ are the coefficients obtained from kernel ridge-regression (KRR). This suggests the following two stage algorithm: first, we learn the regularisation constant, $\lambda$, and parameters of the kernel matrix, $K$, to maximize KRR performance on hold-out data; next, we perform variational IB, with fixed $K$ and $\lambda$.

## 4.1 Simulations

To illustrate the capabilities of the kernel IB algorithm, we considered an 'occlusion' task, with the outer columns of each patch presented as input, $X$ (2 columns to the far left and right), and the inner columns as the relevance variable $Y$, to be reconstructed. Image patches were as before. Note that performing the occlusion task optimally requires detecting combinations of features presented to either side of the occluded region, and is thus inherently nonlinear.

We used gaussian kernels, with scale parameter, $\kappa$, and regularisation constant, $\lambda$, chosen to maximize KRR performance on test data. Both test and training data consisted of 10,000 images. However, $A$ was restricted to lie on a subset of 1000 randomly chosen training patches (see earlier).

Figure 3B shows a selection of decoding filters ($U$) learned by the sparse kernel IB algorithm ('sparse kIB'). A large fraction of filters resembled near-horizontal bars, traversing the occluded region. This was not the case for the sparse linear IB algorithm, which recovered localized blobs either side of the occluded region, nor the gaussian linear or kernelized models, which recovered non-local features (fig. 3C). Figure 3D shows a small but significant improvement in performance for the sparse kIB versus the gaussian kIB model. Most noticeable, however, is the distribution of responses, which are much more heavy tailed for the sparse kIB algorithm (fig. 3E).

To demonstrate the non-linear behaviour of the sparse kIB model, we presented bar segments: first to either side of the occluded patch, then to both sides simultaneously. When bar segments were presented to both sides simultaneously, the sparse KIB model 'filled in' the missing bar segment,

in contrast to the reconstruction obtained with single bar segments (fig. 3F). This behaviour was reflected in the non-linear responses of certain encoding units, which were large when two segments were presented together, but near zero when one segment was presented alone (fig. 3G).

Finally, we repeated the occlusion task with handwritten digits, taken from the USPS dataset (`www.gaussianprocess.org/gpml/data`). We used 4649 training and 4649 test patches, of $16 \times 16$ pixels. However, expansion coeffecients were restricted to a lie on subset of 500 randomly patches. We set $X$ and $Y$, to be the left and right side of each patch, respectively (fig. 4A).

In common with the artificial data, the response distributions achieved with the sparse kIB algorithm were more heavy-tailed than for the gaussian kIB algorithm (fig. 4B). Likewise, recovered decoding filters closely resembled handwritten digits, and extended far into the occluded region (fig. 4C). This was not the case for the alternative IB algorithms (fig. 4D).

## 5   Discussion

Previous work has shown close parallels between the IB framework and maximum-likelihood estimation in a latent variable model [12, 13]. For the sparse IB algorithm presented here, maximizing the IB objective function is closely related to maximizing the likelihood of a 'sparse coding' latent variable model, with student-t prior and linear gaussian likelihood function. However, unlike traditional sparse coding models, the encoding (or 'recognition') model $p(r|x)$ is conditioned on a seperate set of inputs, $X$, distinct from the image patches themselves. Thus, the solutions depend on the relation between $X$ and $Y$, not just the image statistics (e.g. see fig. 2). Second, an additional parameter, $\gamma$, not present in sparse coding models, controls the trade-off between encoding and compression. Finally, in contrast to traditional sparse coding algorithms, IB gives an unambiguous ordering of features, which can be arranged according to the response variance of each unit (fig. 1F).

Our work is also closely related to the IM algorithm, proposed by Barber et al. to solve the information maximization ('infomax') problem [14]. However, a general issue with infomax problems is that they are usually ill-posed, necessitating additional *ad hoc* constraints on the encoding weights or responses [15]. In contrast, in the IB approach, such constraints emerge automatically from the bottleneck term.

A related method to find low-dimensional projections of $X$/$Y$ pairs is canonical correlation analysis ('CCA'), and its kernel analogue [16]. In fact, the features obtained with gaussian IB are identical to those obtained with CCA [6]. However, unlike CCA, the number and 'scale' of the features are not specified in advance, but determined by the bottleneck parameter, $\gamma$. Secondly, kernel CCA is symmetric in $X$ and $Y$, and thus performs nonlinear embedding of both $X$ *and* $Y$. In contrast, the IB problem is assymetric: we are interested in recovering $Y$ from an input $X$. Thus, only $X$ is kernelized, while the decoder remains linear. Finally, the features obtained from gaussian IB (and thus, CCA) differ qualitatively from the sparse IB algorithm, which recovers sparse features that account jointly for $X$ and $Y$.

Sparse IB can be extended to the nonlinear regime using a kernel expansion. For the gaussian model, the expansion coefficients, $A$, are a linear projection of the coefficients used for kernel-ridge-regression ('KRR'). A general disadvantage of KRR, is that it can be difficult to know which aspects of $X$ are relied on to perform the regression. In contrast, the kernel IB framework provides an intermediate representation, allowing one to visualize the features that jointly account for both $X$ and $Y$ (figs. 3B & 4C). Furthermore, this learned representation permits generalisation across different tasks that rely on the same set of latent features; something not possible with KRR.

Finally, the IB approach has important implications for models of early sensory processing [17, 18]. Notably, 'efficient coding' models typically consider the low-noise limit, where the goal is to reduce the neural response redundancy [7]. In contrast, the IB approach provides a natural way to explore the family of solutions that emerge as one varies internal coding constraints (by varying $\gamma$) and external constraints (by varying the input, $X$) [19, 20]. Further, our simulations suggest how the framework can be used to go beyond early sensory processing: for example to explain higher-level cognitive phenomena such as perceptual filling in (fig. 3G). In future, it would be interesting to explore how the IB framework can be used to extend the efficient coding theory, by accounting for modulations in sensory processing that occur due to changing task demands (i.e. via changes to the relevance variable, $Y$), rather than just the input statistics ($X$).

## Footnotes

[1] $f(\nu_i, \xi_i, a_i) = \log\Gamma\left(\frac{\nu_i}{2}\right) - \frac{\nu_i}{2}\log\frac{\nu_i}{2} - \frac{1}{N}\sum_n \left[\frac{\nu_i - 1}{2}\left(\psi(a_i) - \ln\frac{a_i}{\xi_{ni}}\right) - \frac{\nu_i}{2}\xi_{ni} + H_{ni}\right]$, where $H_{ni}$ is the entropy of a gamma distribution with shape and rate parameters: $a_i$, and $a_i/\xi_{ni}$, respectively [9].

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
