[Reviews · NeurIPS 2016]

Reviewer 1

Summary

The paper combines two concepts that gained significant interest in theoretical neuroscience: information bottleneck (IB) and sparse coding. As such, in Reviewer's opinion, it has a potential of making a big impact on the field of neural coding. The paper is technically sound, proposing a number of algorithms: a EM-IB type of algorithm assuming sparse approximate prior p(encoding variable), and a kernel extension to the algorithm, which allows to deal with more complex, non-linear relationships between observables. It analyses a number of interesting problems. Perhaps, the writeup would benefit from underlining its neuroscientific/neural coding aspect (i.e. by relating results also to those obtained by a standard sparse coding approach).

Qualitative Assessment

Thank you for the very interesting read. I enjoyed the comprehensive description of all the concepts and methods and, as stated above, I hope this research will make a big impact on theoretical neuroscience. I have a few questions rather than comments, and they are mostly to satisfy curiosity. 1. Would you be in a position to hypothesize, why sparse IB filters in Fig. 1C are curved? From your description I understood the stimuli were oriented bars, rather than curved arcs? 2. The horizontal bias in orientation representation - was it observed only with a strong bottleneck? Would the effect go away with a softer constraint on coding? To what extent did sparsity contribute to that result? In fact, there is a large body of work in neuroscience concerned with an asymmetry of orientation representation in V1, the famous "oblique effect" - which looks like an ideal setup for your experiment. I believe you can contribute with an interesting functional hypothesis of how the effect comes about in the brain of highly visual animals. This would probably even be accepted as a separate publication in a more neuroscience-oriented venue. Good luck! The paper is very well written. A few minor typos in lines: 90 (decide between mu or 0), 93 (typo in Xi variable), 95 (row rather than a column?), 175 (features presented on...), 195, 222 (asymmetric), 239 (Fig. 3F-G), 256 (neural) and Equation 1 (X and Y swapped in the first line). Please, double check the X-axis label in Fig. 1F - shouldn't it read "units"? You may consider mentioning the phi(x) [line 146 and following] is a row vector, I got slightly confused by notations such as phi(x) phi^T(x), which I understand should represent dot-products.

Confidence in this Review

2-Confident (read it all; understood it all reasonably well)


Reviewer 2

Summary

The paper extends Gaussian IB to the case of sparse variables, which makes more sense for data that have sparse or nonlinear manifold like structure. Performance is demonstrated on some toy image patch data, in addition to hand written digits, showing that the algorithm can discover relevant structure in the data.

Qualitative Assessment

Overall this is an interesting paper. However it would have been nice to see some applications to real data such as natural images rather than the toy image patch data. Also this sentence from the discussion is confusing: 'However, unlike traditional sparse coding models, the encoding (or ‘recognition’) model p(r|x) is conditioned on a seperate set of inputs, X, distinct from the image patches themselves. Thus, the solutions depend on the relation between X and Y , not just the image statistics. 
' Specifically, it is not clear what is meant by 'a separate set of inputs, X, distinct from the image patches themselves.' My understanding is that Y is just the image to be reconstructed, so this seems very much like sparse coding. Specific comments: Line 39 says that equation 1 is being maximized. I believe it should be minimized. That is at least what is shown in Tishby's paper, eq. 15. The error is also repeated on line 52. Figure 1 example - why not use natural images? would have been more compelling example Occlusion example with kernel - a generative model could also do this, how does this method compare?

Confidence in this Review

2-Confident (read it all; understood it all reasonably well)


Reviewer 3

Summary

This paper presents an interesting approximation to make the information bottleneck method practical in high dimensional scenarios, which is illustrated with experimental examples. Moreover, since this technique depends on scalar products, authors propose a nonlinear generalization based on kernels. As opposed to other kernel-based feature extraction methods, the proposed kernel-information-bottleneck method has the advantage of getting an intermediate representation where features may be visualized. I think this successful combination (simplified information bottleneck and kernel generalization) are worth publishing at NIPS!

Qualitative Assessment

After a convincing exposition of the proposed simplification of the information bottleneck and its kernel-based nonlinear extension, the authors make an interesing point in the discussion: as opposed to other feature extraction techniques, which usually depend only on the statistics of the input signal, "x", their information-bottleneck features also depend on the relevant signal "y". It would be interesting to see examples of the practical effect of such connection: how the sparse features (e.g. Gabor-like edge detectors in natural images) change if the signals to retain information from are a specific class of images (e.g. faces...)?. May this be connected with top-down adaptation of early vision mechanisms?

Confidence in this Review

2-Confident (read it all; understood it all reasonably well)


Reviewer 4

Summary

This paper develops a method to impose sparse prior to learn encoders and decoders in the information bottleneck framework. The authors show how one can use variational approximation to solve the optimization problem. The method is shown to learn Gabor-like filters on image patches, similar to those learned from traditional sparse coding. The authors further describe a kernel method extension which allows one to learn non-linear encoders. The effectiveness is demonstrated through an image inpainting task of occluded hand-written digits.

Qualitative Assessment

This paper is well-written and the methods are sound. It is interesting to see how one can incorporate sparse prior in the information bottleneck framework, and yield visually similar results to conventional sparse coding. However, the effectiveness of the sparse prior is well-known and there already exist various formulations that use the idea of sparsity (e.g., LASSO, sparse autoencoder, etc). The paper will be much stronger if the authors can provide more insights and quantitative comparisons to justify the benefits from the IB method. The method seems particularly related to sparse auto-encoder in that both methods can yield linear encoder and decoder. The experiments reported seem a bit too simple to me, mostly performed on 9x9 image patches without further analyzing the results. Hence it is unclear in what circumstances one would consider the IB method. The example applications shown in this paper, image denoising and image inpainting, both can be solved with conventional sparse coding. Is the IB method better in these tasks? Overall, I think this is a sound paper, but more comparisons with existing methods are needed to justify its value. Detailed comments: 1. Line 32 mentions varying gamma to trade off between encoding quality and compression. This can be similar done in other sparse coding methods, for example, the strength of l1-norm regularization in LASSO. I don't see this as a unique quality of sparse IB. 2. The other difference to conventional sparse coding is that the IB method learns particular input-output relationship. How does one choose a proper output? 3. Line 73: Do you mean q(y|r) = N(y|Ur)?

Confidence in this Review

2-Confident (read it all; understood it all reasonably well)


Reviewer 5

Summary

This paper combines the information bottleneck method (IB) approach with sparse coding. It proposes two instantiations of IB: first one is using sparse priors on the internal representation, and the second one is a kernel extension of the first model. A variational algorithm for learning the model is proposed, and the model is evaluated on simulated and real data (handwritten digits).

Qualitative Assessment

I find the paper novel and interesting. To my knowledge the algorithm is original and it adds to the existing tollbox of IB based approaches. The proposed method seems to outperform Gaussian IB on denoising and occlusion/inpaiting tasks on simulated and real data. It also provides new analysis tools for sparse representations in the form of IB information curves. Overall I think this work has many promising applications in machine learning and neuroscience and would be of interest to the NIPS audience. Below I list a few questions / suggestions which could help authors to further improve the quality/readability of the paper: 1) As I understand, encoding/decoding dictionaries $W, U$ are trained for a fixed constraint $\gamma$. What is the influence of $\gamma$ on trained dictionaries? Do different receptive fields emerge under different constraints? 2) The authors validate the model using simulated data and handwritten digits. While these are necessary tests which demonstrate the correctness of the method, used datasets are somewhat constrained. It would be interesting to see the algorithm validated on a more diverse set of stimuli e.g. natural image patches. Alternatively, I think that the authors should provide a brief justification for their choice of data (handwritten digits). 3) Perhaps it would be good to include a brief justification for the choice of the sparse prior. Why did the authors choose Student-t distribution? Is it analytical convenience, or are there some other reasons? Why not a generalized Gaussian for instance? Different priors have different entropies, which as I understand should affect the performance of the model. 4) I like the discussion which relates the proposed model to previous infomax/sparse coding approaches. Can we understand them as special cases of the sparse IB model? Minor comments: While overall the paper is very clearly written I have a feeling that in a few cases an additional line of explanation would substantially improve reception. For instance in equation (5) - it is not immediately obvious what is the relationship between the upper bound on I(X;R) and the log-likelihood of R. Line 38 - the objective is to minimize the Lagrangian L (in the form the authors present it), not maximize it Line 44 - I do not understand that sentence - what does "and" refer to?, Also - the inequality within the sentence could be given some more context for clarity. Figure 2 D - as I understand "stimuli" on that panel means Y, not X. Perhaps you could clarify that.

Confidence in this Review

2-Confident (read it all; understood it all reasonably well)


Reviewer 6

Summary

The information bottleneck method seeks to extract the maximum information from variable X about variable Y while minimizing the superfluous information. The practical application of the method is hindered by the computational demands of analyzing high-dimensional or non-Gaussian data. The paper proposes an alternative objective that provides a lower bound on the information bottleneck objective while being more tractable. The paper provides an algorithm based on this objective and demonstrates its application in two examples. Finally, they provide a kernel version of the method and apply it to an occlusion task

Qualitative Assessment

The paper provides a clear description of its method. The detailed analysis of the simple case, the analysis of the effect of an variation, and the application to a task provide the reader with an understanding of the performance and potential applications of the method. The figures are clear and their captions allow the reader to get a basic understanding of what they intend to show without needing to search for the relevant sections of the text. I expect that people will be able to put this method to productive use.

Confidence in this Review

2-Confident (read it all; understood it all reasonably well)